# Time-Resolved Kinetic Measurement of Microalgae Agglomeration for Screening of Polysaccharides-Based Coagulants/Flocculants

**DOI:** 10.3390/ijerph192114610

**Published:** 2022-11-07

**Authors:** Jinxia Zhou, Yunlu Jia, Xiaobei Gong, Hao Liu, Chengwu Sun

**Affiliations:** 1State Key Laboratory of Pulp and Paper Engineering, South China University of Technology (SCUT), Guangzhou 510640, China; 2State Key Laboratory of Freshwater Ecology and Biotechnology, Institute of Hydrobiology, Chinese Academy of Sciences, Wuhan 430072, China; 3Bengbu-SCUT Research Center for Advanced Manufacturing of Biomaterials, Bengbu 233010, China

**Keywords:** time-resolved measurement, microalgae, coagulants/flocculants (CFs), filtering-flowing analysis (FFA), kinetics model, cationic polysaccharides, bacterial exopolysaccharides (BEPS)

## Abstract

Time-resolved monitoring of microalgae agglomeration facilitates screening of coagulants/flocculants (CFs) from numerous biopolymer candidates. Herein, a filtering-flowing analysis (FFA) apparatus was developed in which dispersed microalgal cells were separated from coagulates and flocs formed by CFs and pumped into spectrophotometer for real-time quantification. Polysaccharides-based CFs for *Microcystis aeruginosa* and several other microalgae were tested. Cationic hydroxyethyl cellulose (CHEC), chitosan quaternary ammonium (CQA) and cationic guar gum (CGG) all triggered coagulation obeying a pseudo-second-order model. Maximal coagulation efficiencies were achieved at their respective critical dosages, i.e., 0.086 g/g*_M.a._* CHEC, 0.022 g/g*_M.a._* CQA, and 0.216 g/g*_M.a._* CGG. Although not active independently, bacterial exopolysaccharides (BEPS) aided coagulation of *M. aeruginosa* and allowed near 100% flocculation efficiency when 0.115 g/g*_M.a._* CQA and 1.44 g/g*_M.a._* xanthan were applied simultaneously. The apparatus is applicable to other microalgae species including *Spirulina platensis*, *S. maxima*, *Chlorella vulgaris* and *Isochrysis galbana*. Bio-based CFs sorted out using this apparatus could help develop cleaner processes for both remediation of harmful cyanobacterial blooms and microalgae-based biorefineries.

## 1. Introduction

Coagulation/flocculation process (CFP) is a prevalent pretreatment technology for efficient removal of harmful cyanobacteria blooms (HCBs) from water bodies [1]. *Microcystis aeruginosa*, a unicellular prokaryotic cyanobacterium, is one of the most pervasive and hazardous HCBs species [2]. Its small colonies (30~100 μm) are vertically distributed in water [3] and require coagulants (e.g., aluminum salts) together with flocculant aids (e.g., polyacrylamide) to enhance agglomeration [4]. CFP has also been applied to facilitate the dewatering of microalgal biomass from dilute suspensions through cell aggregation and flocs formation [5]. Subsequently, biomass slurry (~7% dryness) is collected by sieving for further drainage and bioproduct extraction [5]. However, the use of chemical coagulants/flocculants (CFs) poses serious environmental and health risks, especially the release of neurotoxic and carcinogenic substances [6]. Moreover, chemical CFs may cause operation problems in downstream biorefining processes, such as inactivation of microbes and enzymes, introduction of organic impurities, accumulation of non-biodegradable sludge [7].

Bio-based CFs are environmental-friendly macromolecular materials derived from plants [8], animals [9], fungi [10], or bacteria [11]. Modified tannin, chitosan, fungal chitin and bacterial exopolysaccharides (BEPS), have been investigated for sustainable HCBs removal [8,9,10,11]. Applicability of bio-based CFs has also been verified in recovery of cell biomass from growth cultures of *Spirulina platensis* [12], *Chlorella vulgaris* [13] and *Isochrysis galbana* [14]. Since different microalgae species have different cell shapes, sizes and surface charges, it is important to select and develop universal bio-based CFs from numerous biosources. It is also desired to search for selective biocoagulants that could separate harmful species, e.g., *M. aeruginosa*, from contaminated microalgal cultures. Every year many new CFs-producing strains are reported, providing us with abundant candidates for experimental trials. However, it is very laborious and time-consuming to perform the screening work involving coagulation, filtration, drying and weighing step by step, especially when studying combinations of different CFs.

Time-resolved (TR) monitoring of dispersed cells during CFP of microalgae could facilitate the screening process. Rapid and automatic acquisition of data supports further rational design of CFs’ structures as well as formulation optimization. The high-resolution kinetic profiles obtained could further reveal the correlation between structural features of CFs and their performance in CFP of microalgae. This is beneficial to mechanism elucidation of novel CFs candidates. Many TR methods have been reported for studying microalgae. For instance, a TR fluoroimmunometric assay was established for detection of microcystins [15]. TR microfluorimetry was developed for free radical and metabolic rate detection in microalgae [16]. TR ICP-MS was employed for simultaneous cell counting and determination of constituent metals in microalgal cells [17]. This paper reports a filtering-flowing analysis (FFA) apparatus enabling TR measurement of microalgae agglomeration for screening effective bio-based CFs. Polysaccharide-based CFs (PBCFs) were tested, and the kinetics properties were investigated toward *M. aeruginosa* and four other microalgal species.

## 2. Materials and Methods

Cationic guar gum (CGG) and chitosan quaternary ammonium (CQA) salt were purchased from Tianjin Baima Technology Co., Ltd. (Tianjin, China). Cationic hydroxyethyl cellulose (CHEC) was prepared following Wang and Ye’s method [18]. Xanthan was purchased from Shandong Yousuo Chemical Technology Co., Ltd. (Shandong, China). Two other bacterial exopolysaccharides (BEPS) were produced by *Paenibacillus mucilaginosus* (GDMCC No. 62049) and *Agrobacterium* sp. (GDMCC No. 62125), respectively. These two strains were isolated in our lab and preserved in Guangdong Microbial Culture Collection Center (Guangdong, China). Both strains were cultivated, respectively, in medium containing sucrose 5 g/L, Na_2_HPO_4_ 2 g/L, MgSO_4_ 0.5 g/L, CaCO_3_ 0.1 g/L, and FeCl_3_ 0.005 g/L. After incubation at 37 °C for 3 days, the culture broth was centrifuged (12,000 rpm) at 4 °C for 10 min. BEPS in supernatant was precipitated with absolute ethanol. The precipitate was re-dissolved and centrifuged to remove insoluble particles. The supernatant was re-precipitated with ethanol. The finally obtained precipitate was freeze-dried to produce purified BEPS. Details of these PBCFs including CHEC, CGG, CQA, xanthan, *Agrobacterium* mucopolysaccharides (AMP) and *P. mucilaginosus* exopolysaccharides (PmEPS) are summarized in Appendix A.

*M. aeruginosa* and four other microalgae species were all grown in BG11 medium (Haibo Biotech. Co., Ltd., Shangdong, China) at 25 °C, 2000 Lux for 7 days. The growth cultures were properly diluted with sterilized water to a certain optical density before use. PBCFs (0.5 mL) were added in 50 mL of microalgal culture broth followed with magnetic stirring to start CFP. The mixture suspension was filtered with a 30-mesh sieve tube and pumped into a flowing quartz cuvette (optical path 1 cm) for quantification of microalgal cells. Data were recorded every 1 s to produce a TR profile for further kinetics analysis. The variation in cell concentration (Δc, g/L) was calculated from the Lambert-Beer law (Equation (1)). 

The coagulation number of microalgae at t min (Q_t_, g_m_) and adsorption capacity (q_t_, g_m_/g_CFs_) of polysaccharides-based CFs were deduced from Δc (Equations (2) and (3)). The unit g_m_ means grams of dried microalgal cells and g_m_/g_CFs_ represents grams of dried microalgal cells coagulated by per gram of CFs. Specifically, g*_M.a_*. and g*_C.v._* are used to replace g_m_ for expressing the dry mass of *M. aeruginosa* and *C. vulgaris*, respectively. Flocculation efficiency (FE, %) was calculated from the coagulation amount at equilibrium (Q_e_, g_m_) according to Equation (4).
(1)$$\Delta c=\frac{A_{\lambda }^{0}-A_{\lambda }^{t}}{\varepsilon }$$
(2)$$Q_{t}=\Delta c\cdot V$$
(3)$$q_{t}=\frac{Q_{t}}{m_{\mathrm{CFs}}}$$
(4)$$\mathrm{FE}=\frac{q_{t}\cdot m_{\mathrm{CFs}}\cdot \varepsilon }{A_{\lambda }^{0}\cdot l\cdot V}\times 100\%$$
where, $A_{\lambda }^{0}$ and $A_{\lambda }^{t}$ are absorbance at an identical wavelength at time 0 and t, u.a.; ε represents absorption coefficient, L/g/cm; V is the volume of microalgae suspension, L; m_CFs_ is the dry mass of bio-based CFs, g_CFs_; l is the path length of cuvette, cm.

All kinetics plots were fitted with pseudo-first-order (or Lagergren equation, Equation (5)), pseudo-second-order (Equation (6)) and Weber and Morris models (Equation (7)) using Origin 9.0 [19].
(5)$$\ln\;\left(q_{e}-q_{t}\right)={\;\ln\;q}_{e}-k_{1}t$$
(6)$$\frac{t}{q_{t}}=\frac{1}{k_{s}q_{e}^{2}}+\frac{t}{q_{e}}$$
(7)$$q_{t}=k_{\mathrm{ip}}t^{\frac{1}{2}}+C$$
where q_e_ (g_m_/g_CFs_) and q_t_ (g_m_/g_CFs_) are the adsorption capacities in the equilibrium state and at time of t, respectively; k_1_ (1/min), k_s_ (g/g/min) and k_ip_ (g/g/min^1/2^) are the rate constants in Equations (5)–(7), respectively; C is a constant related to thickness and boundary layer.

## 3. Results

### 3.1. Quantification of Microalgae by Visible Spectroscopy

All microalgae strains investigated including *M. aeruginosa*, *S. platensis, S. maxima, C. vulgaris* and *I. galbana* had an identical absorbance peak at 680 nm in culture broths (Figure 1a). Lambert–Beer Law is valid for all of them within the cell concentration range of 0.03~0.3 g/L, on the basis of oven-dried mass (Appendix A). Their respective absorption coefficients (ε) are summarized in the legend box in Figure 1a. The microalgal pigments are all intracellular components [20] because neither supernatants nor filtrates of cultures had detectable A_680_ signals. Therefore, there was no colorful extracellular metabolite that interfered with the quantification of microalgal cells by visible spectroscopy. The stability of data output was tested in our FFA apparatus since stable colloidal particles of microalgal cells may be destabilized by shear stress [21]. Figure 1b shows all cell particles of *M. aeruginosa*, *C. vulgaris* and *I. galbana* completely penetrated the holes of filter (30 mesh, or 600 μm of hole size), producing steady consistent absorption lines along with time up to 10 min. A small deviation within ±0.0005 u.a. represents the systematic error range of this apparatus. Obviously, shear stress due to magnetic stirring, filtering, and fluid flowing did not influence the TR coagulation measurement of the three microalgae. Profiles of *S. platensis* and *S. maxima* showed slight decline trend with a slope of −0.0243 and −0.0201 u.a./min caused by 1 g/L of cells, respectively. Considering their larger size dimensions (Appendix A) than the other three microalgae species, such instability could be mainly attributed to self-coagulation of cell particles rather than interception by the filter or other reasons [22]. These decline factors should be taken into consideration in subsequent coagulation measurements and kinetics calculation. In addition, potential interference from optical absorption by PBCFs was also evaluated (Appendix A). All the tested PBCFs at a concentration of up to 200 mg/L had spectra (400~1000 nm) overlapped with the control line. In summary, the FFA apparatus enables stable output of data during the CFP of microalgae.

### 3.2. TR Kinetic Coagulation Profiles and Model Fitting Curves

Overall TR profiles of A_680_ variation were recorded for *M. aeruginosa* and other microalgal species. The FFA apparatus was started up with magnetic stirring (stirrer diameter 5 mm, length 15 mm, speed 800 rpm) and continuous flow circulation of microalgal suspension (20 s for a cycle; 10 mL of dead volume) in a 200 mL flask. After resisting for 15 s (delay time for checking stability), a certain number of cationic polysaccharides, e.g., CHEC, were pipetted to initiate coagulation (See the dosing timepoint in Figure 2a). Decrease in A_680_ was first recorded at around 45 s because the initial mixture with coagulates filtered out could only be detected after filling the cuvette cell in spectrophotometer from the stirring flask.

The obtained TR profiles can be roughly divided into two phases: fast (logarithmic) and slow (linear). The curve-to-tangent (CT) point of the two phases or the time approaching quasi-stable state was determined by the dosage and species of cationic polysaccharides. Figure 2a also demonstrated good data reproducibility from triplicated independent measurements. Standard deviation of A_680_ at any point in the fast and slow phase was within ±0.009 and ±0.003 u.a., respectively. Therefore, minor changes in *M. aeruginosa* concentration over ±2.83 mg/L (0.142 mg in the 50 mL test flask) could be sensitively detected.

CFP curve plots (Figure 2b) were generated from the TR profiles by deducing q_t_ from A_680_ according to Equations (1) and (3). The adsorption capacity, q_t_, represents the coagulation performance of cationic polysaccharides per unit mass. Figure 2b shows the q_t_ curve became lower when CHEC dosage was promoted from 0.014 to 0.086 g/g*_M.a._* although many more *M. aeruginosa* cells were coagulated in total. As more CHEC (e.g., 0.144 g/g*_M.a._*) was used, a curve showing less efficient coagulation was obtained as demonstrated in Figure 2a. This is probably due to the repulsive interactions among *M. aeruginosa* cell particles bound with excessive positively charged CHEC. After plotting the measured Q_t_ at 9 min (denoted as Q_9′,_ See Figure 3a), it is clear that there was a critical concentration point below which Q_9′_ increased with CHEC dosage, while above that Q_9′_ decreased. Higher CHEC dosage up to 0.288 g/g*_M.a._* led to poorer coagulation.

All the CFP curve plots of *M. aeruginosa* with addition of cationic polysaccharides could be well fitted by pseudo-first-order (R^2^ > 0.94) and pseudo-second-order adsorption kinetics models (R^2^ > 0.99, Figure 2b,c). The measured q_t_ at 9 min (q_9′_) was proportional to the q_e_ deduced from the two models as illustrated in Figure 2d. The q_e_ deduced from the pseudo-first-order model (q_e_^1st^) was apparently lower than q_9′_. The q_e_ values predicted with pseudo-second-order model (q_e_^2nd^) were more consistent with the experimental q_9′_ values. The q_e_^2nd^ was 1.03~1.05 folds of q_9′_ while q_e_^1st^ was 0.92~0.94 folds of q_9′_, depending on the selection of cationic polysaccharides (Figure 2d). Obviously, the pseudo-second-order model was more appropriate to describe the TR coagulation curves of *M. aeruginosa*, reflecting chemical adsorption mechanisms such as ion exchange and surface complexation dominate the CFP [23].

Coagulation of *S. platensis*, *S. maxima*, and *C. vulgaris* by CQA also followed the pseudo-second-order model (R^2^ > 0.985, Table 1). Initial coagulation velocities derived from the model k_s_·q_e_^2^ were compared, and data in Table 1 showed *S. maxima* was most readily removed. Among all tested species, *I. galbana* was a particular one that could not be coagulated by any of the studied cationic polysaccharides.

Weber and Morris kinetics model could not describe the whole coagulation curve over 9 min (Appendix A). It predicted well only the CFP data in the initial 4 min with correlation coefficients (R^2^) higher than 0.9. Data in Appendix A showed the highest R^2^ was only 0.946 for the tested specimen. Obviously, the Weber and Morris model was not suitable to describe the coagulation of microalgae by cationic polysaccharides.

BEPS such as AMP, PmEPS and xanthan did not coagulate *M. aeruginosa* or other microalgae investigated in this work even at concentrations as high as 0.5 g/L (7.2 g/g*_M.a._*), as shown in Figure 2c and Appendix A. Since both BEPS and microalgal cells carry negative charges on the surface, Coulombic repulsion forces of the charged cells were reinforced with increase in the strength of the electric field. Obviously, electrostatic neutralization is necessary for achieving coagulation.

### 3.3. CFP Performance of Cationic Polysaccharides toward M. aeruginosa and C. vulgaris

*M. aeruginosa*, a typical prokaryotic bloom-forming cyanobacterium, and *C. vulgaris*, a cell-factory eukaryotic microalgae for biorefinery, were used to further explore the CFP behaviors of cationic polysaccharides. Results in Figure 3 show that Q_e_ predicted by pseudo-second-order model was similar to Q_9′_ and also increased with the dosage of CFs to a critical concentration point above which the coagulation amount was decreased. For *M. aeruginosa*, the maximum values of Q_e_ were 2.60, 2.88 and 1.65 mg*_M.a._*, achieved by 0.086 g/g*_M.a._* CHEC, 0.216 g/g*_M.a._* CGG, and 0.022 g/g*_M.a._* CQA, respectively, which were in consistence with the dosages for maximal Q_9′_. Among them, CQA showed the highest efficiency for coagulation of *M. aeruginosa*, reaching 88.3% flocculation efficiency at a small dosage of 0.022 g/g*_M.a._*.

The coagulation capacities at equilibrium (q_e_) of the three cationic polysaccharides were measured using *M. aeruginosa* and *C. vulgaris* and the results are shown in Figure 3c,d. CQA had a significantly higher coagulation capacity within the dosage range of 0.01~0.05 g/g*_M.a._* than that of CHEC or CGG. A drastic decrease in q_e_ with dosage of CQA suggests that excessive CQA macromolecules led to repulsion instead of agglomeration of neutralized microalgal cells. CHEC exhibited moderate capacity and a downward trend with a lower slope than CQA, while CGG had a relatively flat curve of q_e_ vs. dosage (Figure 3a). Different activities of these coagulants were probably due to their intensive cationic charge densities on surface as expressed by zeta potential (Appendix A).

In the case of *C. vulgaris*, the critical concentration point for CQA was around 0.03 g/g*_C.v._*, lower than that of CHEC or CGG (both around 0.057 g/g*_C.v._*). However, maximal Q_e_ by CQA in Figure 3b was only 6.53 mg*_C.v._* (Q_max_ was 11.71 mg*_C.v._*), less than that by CHEC (7.90 mg*_C.v._*) or CGG (8.71 mg*_C.v._*). CGG had q_e_ at a higher level than CQA or CHEC within the dosage range of 0.04~0.11 g/g*_C.v._* (Figure 3d). On the other hand, CGG in this concentration range had a relatively low CFP efficiency toward *M. aeruginosa* than *C. vulgaris* (Figure 3c). Different performance of these coagulants allowed selective separation of *M. aeruginosa* from *C. vulgaris*. For example, when 0.05 g/g_m_ CGG was applied, *C. vulgaris* could be selectively coagulated from the contaminants of *M. aeruginosa*. 

### 3.4. Performance of BEPS as Flocculation Aids

Our apparatus is also applicable to develop dual-component CFs. Figure 4 shows BEPS including AMP, PmEPS and xanthan could all serve as flocculation aids to enhance the CFP efficiency of *M. aeruginosa* by cationic polysaccharides. Two approaches for addition of BEPS, stepwise or simultaneously with cationic polysaccharides, were studied in the FFA apparatus (Figure 4). Kinetic measurements of A_680_ variation clearly demonstrated that both addition methods led to higher flocculation efficiency (FE) of microalgal cells. Maximal FE approaching 100% was achieved by simultaneous addition of CQA (0.115 g/g*_M.a._*) and xanthan (1.44 g/g*_M.a._*), while stepwise addition of them could remove about 90% *M. aeruginosa* cells through CFP (Figure 4a,b). The contribution ratio of CQA to xanthan was 4:5 as calculated from the A_680_ at 4 min in Figure 4a. In fact, the contribution of xanthan was still dependent on the presence of CQA because a single addition of xanthan did not initiate measurable microalgae agglomeration (Figure 2c and Appendix A). Xanthan in the other two combinations (with CHEC and CGG, repectively) also promoted the flocculation efficiency (Figure 4c), showing its universal applicability as flocculation aids. Similar tends were observed in the combination sets consisting of PmEPS and AMP although they were less efficient than xanthan.

The constant k_s_ and initial coagulation velocity (v_0_) of *M. aeruginosa* with addition of dual-component PBCFs were calculated from data in Figure 4b and are shown in Figure 4d. The addition of BEPS, especially xanthan, decreased the value of k_s_ but increased v_0_, suggesting additional adhesive function of BEPS in cooperation with CQA. Further deep mechanism investigations are necessary to be performed. Our apparatus is applicable to generate precise kinetics data and save laborious batch measurements.

## 4. Discussion

A major function of the FFA apparatus is to provide time-resolved kinetic profiles microalgae coagulation. The three kinetics models used in this study were originally developed for a mathematical description of adsorption process [24]. Coagulation and flocculation are very similar with adsorption of free cell particles onto CFs-attached cell aggregates. These models have been widely used to evaluate the performance of bio-based CFs in microalgae recovery [25,26,27]. Coagulation of microalgae by eggshell [25], fungal pellets [26], and anaerobic sludge-derived BEPS [27] had all been found well fitted with pseudo-second-order kinetics model. However, in these reports, experimental data for the modeling study were obtained by batch sampling and one-by-one measurement [25,26,27]. The fitted curves contained usually less than 10 points, not to mention that the time intervals were generally higher than 5 min in order to reduce the time delay error. The present method provides time-resolved profiles by using the automatic apparatus with high resolution and reliability, significantly saving time and labor.

Marine microalgae have been widely utilized for the production of value-added compounds [28]. Therefore, pH, salts and temperature have to be controlled to simulate their native growth conditions [28]. It is convenient to study the harvest of cell biomass by CFs under varied environmental parameters. Controlled temperatures would be available by placing the stirring flask in a magnetic water bath. Desired pH values and ion strengths could be obtained by addition of chemicals into the reaction flask. 

A spectrophotometer was used as detector in the present apparatus, which allows simultaneously monitoring coagulation of two different colorful microorganisms. For instance, *Serratia marcescens* is a Gram-negative bacillus commonly found in water and soil, capable of producing prodigiosin, an intracellular red pigment [29]. A *S. marcescens* strain, LTH-2, and its pigment were reported having strong *Microcystis*-lysing activity [30]. It is important to study the effect of CFs on the microalgae-bacteria ecosystem. Then it can be done using our apparatus to record the variations in absorbance at the respective specific wavelengths for *S. marcescens* and *M. aeruginosa*. Dual-wavelength calculation methods can be established following Liu et al.’s previous article [31]. A drawback is that this research method seems inapplicable to colorless bacteria. 

Edible microalgae cultures are often contaminated by *M. aeruginosa*. Selective separation of different microalgal species could be very useful in the microalgal biorefinery [32]. Selective adsorbents for the above purpose have been long considered as a technical barrier [33]. Our data in Figure 3 showed a possibility of developing particular bio-based CFs for selective separation of *C. vulgaris* from *M. aeruginosa*. It may help minimize the accumulation of cyanotoxin in bioproducts from *C. vulgaris*. The optimal application techniques including CFs components, dosage, temperature and some other conditions deserve further exploration.

## 5. Conclusions

Our FFA apparatus enables online filtering of coagulates, real-time flowing detection and quantification of microalgae biomass variation. It is applicable to measuring TR profiles of microalgae agglomeration and rapid screening of bio-based coagulants/flocculants. Among all tested candidates, cationic polysaccharides had strong coagulation capacity toward *M. aeruginosa* and four other microalgae species. Their TR coagulation profiles obeyed pseudo-second-order kinetics model. Maximal coagulation of *M. aeruginosa* was achieved at the respective critical dosages which were 0.086 g/g*_M.a._* CHEC, 0.022 g/g*_M.a._* CQA and 0.216 g/g*_M.a._* CGG. Different coagulation performance of these cationic polysaccharides potentially allows selective separation of *M. aeruginosa* from beneficial microalgae, i.e., *C. vulgaris*. Bacterial exopolysaccharides did not coagulate any tested microalgae independently but could aid cationic polysaccharides to promote the flocculation efficiency. Maximal flocculation efficiency approaching 100% was achieved by simultaneous addition of 0.115 g/g*_M.a._* CQA and 1.44 g/g*_M.a._* xanthan. This apparatus is also applicable to develop the dual-component polysaccharides-based coagulants/flocculants for microalgae harvest.

## Figures and Tables

**Figure 1 ijerph-19-14610-f001:**
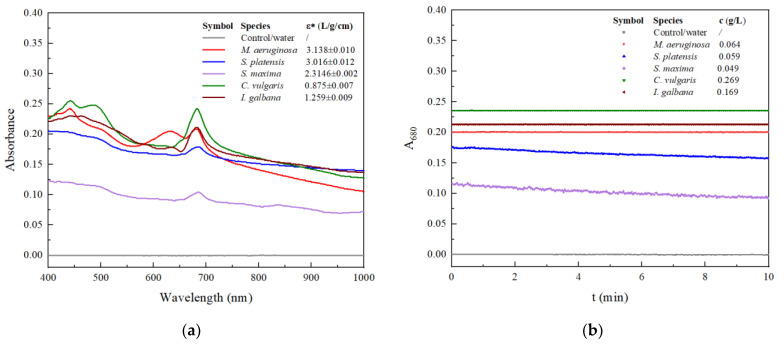
Visible absorption spectra and output data stability for different microalgae species: (**a**) Visible spectra; (**b**) Stability of A_680._ * ε values were reported as the mean ± standard deviation of duplicate independent experiments.

**Figure 2 ijerph-19-14610-f002:**
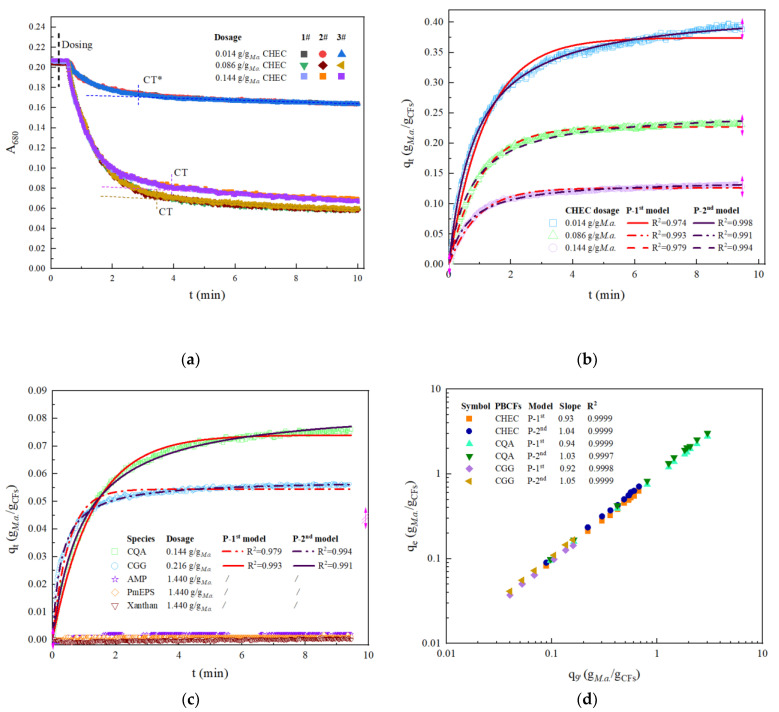
TR kinetic profiles of *M. aeruginosa* coagulation by cationic polysaccharides: (**a**) Triplicated TR variation in A_680_ for CHEC; (**b**) TR coagulation by CHEC fitted with models; (**c**) TR coagulation curves for other PBCFs; (**d**) Linear correlation of q_9′_ and q_e_.

**Figure 3 ijerph-19-14610-f003:**
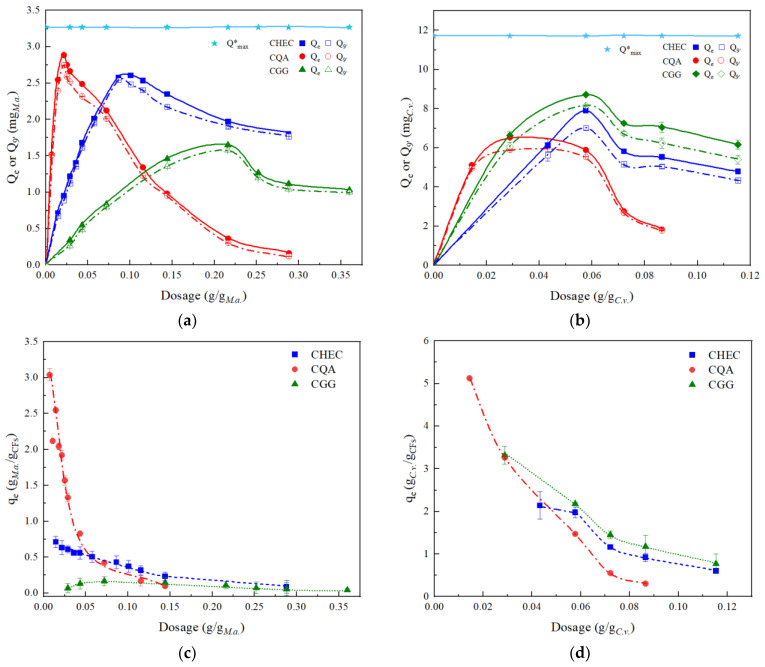
Coagulation amount and adsorption capacity derived from TR kinetic measurements of *M. aeruginosa* and *C. vulgaris*: (**a**) Q_e_ and Q_9′_ for *M. aeruginosa*; (**b**) Q_e_ and Q_9′_ for *C. vulgaris*; (**c**) q_e_ for *M. aeruginosa*; (**d**) q_e_ for *C. vulgaris.* * Q_max_ is theoretical maximum Qe according to the addition amount.

**Figure 4 ijerph-19-14610-f004:**
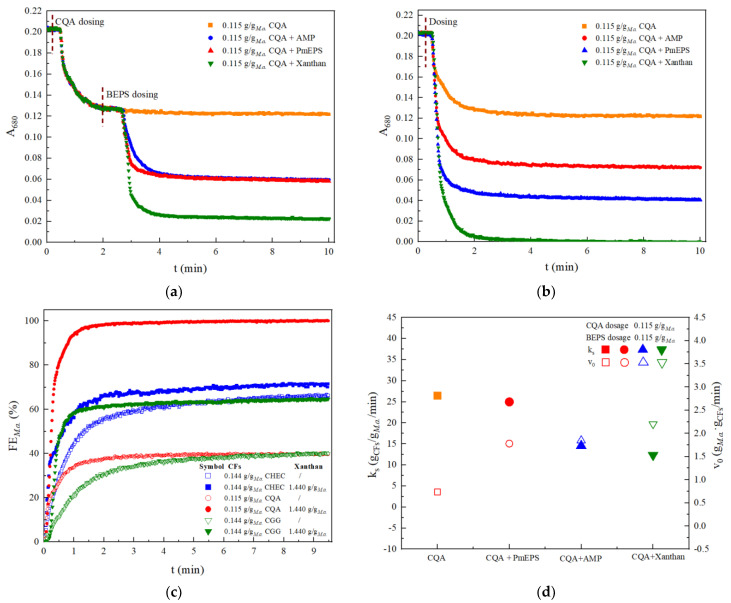
TR kinetic profiles of *M. aeruginosa* coagulation by PBCFs combinations: (**a**) Stepwise addition; (**b**) Simultaneous addition; (**c**) Flocculation efficiency; (**d**) k_s_ and v_0_ deduced from pseudo-second-order model.

**Table 1 ijerph-19-14610-t001:** Parameters of pseudo-second-order kinetics models for comparison of different microalgae.

Microalgae	PBCFs	Dosage (g_CFs_/g_m_)	k_s_ (g_CFs_/g_m_/min)	q_e_ (g_m_/g_CFs_)	k_s_·q_e_^2^ (g_m_·g_CFs_/min)	R^2^
*M. aeruginosa*	CHEC	0.07	3.23	0.437	0.60	0.9906
	CQA	0.07	5.13	0.423	0.93	0.9902
	CGG	0.07	4.92	0.16	0.13	0.9897
*S. platensis*	CQA	0.07	10.08	0.44	1.97	0.9895
*S. maxima*	CQA	0.07	11.11	0.43	2.64	0.9863
*C. vulgaris*	CQA	0.07	128.51	0.06	0.39	0.0989
*I. galbana*	CQA	0.07	Not coagulated			

## Data Availability

Data available on request from the authors.

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
