# Peer review of "Time-Resolved Kinetic Measurement of Microalgae Agglomeration for Screening of Polysaccharides-Based Coagulants/Flocculants"

_ijerph, 2022, doi:10.3390/ijerph192114610_

Round 1
Reviewer 1 Report
The presented results are very interesting. This manuscript will meet the requirements of a research paper with minor revision. I suggest that the Authors divide the results and discussion section into two separate chapters. In my opinion, the authors devoted too little attention to explaining the obtained results, which should be supported by world-class literature. There is a need to broaden the "discussion" section, which is based on too poor literature. After the amendments have been introduced, I believe that the work will gain in value.
Reviewer 2 Report
Kindly find the attachment

Reviewer 3 Report
The manuscript entitled " Time-Resolved Kinetic Measurement of Microcystis aeruginosa Removal for Screening of Polysaccharides-Based Coagulants/ Flocculants " was submitted to the Journal of Environmental Research and Public Health. In this study, the author used a versatile kinetic method to quickly screen effective bio-based coagulants/flocculants, specifically, the bacteria-based flocculants. The topic is interesting and relevant to the scope of this journal. But, among a number of relevant studies, little novelty is found regarding the research contents, methodologies, and findings. This paper needs to revise carefully in many places before it can be considered in the journal. Some suggestions for further improvement are as follows:
1. This paper also did some work on several different algae, so the title should include microalgae, rather than Microcystis.
2. The abstract request completely polish and need a generalized sentence.
3. figures should be self-explainable without reading the main text.
4. Some of the equations in the Results and Discussion are strongly suggested to move into the Method and Materials section.
5. In the process of screening highly efficient flocculant by kinetic method. What is the basis for choosing Microcystis aeruginosa and Chlorella vulgaris.
6. Please explain the meaning of gma, gCFs, gM.a., gC.v..
7. Line 190, in the section of CFP performance of cationic polysaccharides toward M. aeruginosa and C. vulgaris, it is recommended to quote more literatures for discussion.
8. Please pay attention to language problems, especially some proper nouns.
9. It is necessary to briefly describe the advantages of kinetic method for screening flocculants.
